# Korean Consumers’ Recognition of Risks Depending on the Provision of Safety Information for Chemical Products

**DOI:** 10.3390/ijerph17041207

**Published:** 2020-02-13

**Authors:** Seol-A Kwon, Hyun-Jung Yoo, Eugene Song

**Affiliations:** 1National Crisis and Emergency Management Research Institute, Chungbuk National University, Cheongju 28644, Korea; Seolakwon@chungbuk.ac.kr; 2Department of Consumer Studies, Chungbuk National University, Cheongju 28644, Korea; yoohj@chungbuk.ac.kr

**Keywords:** consumer product safety, chemical safety, risk reduction behavior, chemophobia, health knowledge, attitudes, practice, social responsibility

## Abstract

After the 2011 “Oxy accident” involving deaths from humidifier disinfectants, Korean consumers’ anxiety about chemical products has risen. To provide timely, appropriate information to consumers, we must understand their risk recognition and explore methods of safety information provision. We investigated Korean consumers’ level of risk perception for chemical products depending on the provision of safety information and other factors. We conducted an online survey for 10 days with 600 adult Korean consumer participants and analyzed seven factors: catastrophic potential, controllability, familiarity, fear, scientific knowledge, and risk for future generations. Our results showed that married women over 30 perceived chemical products as higher risk, but when information was provided on how to use products safely, catastrophic potential, controllability, fear, scientific knowledge, as well as risk perception, increased significantly. When only risk diagnosis information was provided, catastrophic potential, fear, and risk for future generations remained static, but familiarity had a negative effect (R^2 = 0.586). Age and scientific knowledge affected the recognition of risk when safe risk management methods were provided (R^2 = 0.587). Risk controllability did not have any effect on risk perception. These results suggest that providing information about avoiding or dealing with risks has a positive effect on consumers’ risk perception.

## 1. Introduction

As science and technology continue to develop, consumer goods, including home appliances, household goods, and foods, are becoming increasingly specialized. The problem is that consumer goods containing heavy metals, residual pesticides, food additives, or traces of antibiotics in animal products act as potential risks for consumer safety [1,2,3,4].

Recently in South Korea, consumers’ anxiety about and distrust of chemical products have risen to such an extent that a new word, *chemophobia*, has been coined. Behind this fear is a 2011 incident referred to as “the worst biocide incident” [5,6], in which humidifier sterilizers containing a toxic substance caused many deaths among pregnant women and young children from lung disease and asthma. As of 28 October 2019, a total of 6600 consumers had applied for damage relief, and 1450 people had died [7]. The same hazardous ingredient has been found in household goods such as toothpaste and mouthwash. Accordingly, consumer anxiety about chemical products has been increasing.

Human diseases caused by chemical toxicity are often associated with long-term exposure to chemical products, which can negatively impact human health. Thus, consumers must be more aware of the dangers when using chemical products [8,9]. However, it is very difficult for consumers to determine by themselves the risks involved in products. The provision of risk information is crucial because consumers evaluate and perceive risks through the opinions of experts or government announcements.

Through information and communication technology, the general public receives information in a variety of ways, but people evaluate the same risks differently depending on the amount and content of risk information conveyed through the media. Unlike experts who assess risks based on scientifically proven facts, consumers assess risks subjectively through their own experience, knowledge, and risk information gained from various sources [10]. Safety information interventions change consumers’ attitudes toward safe behaviors as well as their actual behaviors, thus motivating them to process and use health information. Consumers’ safety assessments vary, depending on the quality of information [11,12]. Consumer information affects consumers’ perceptions, and newly provided information changes their perceptions [13,14,15]. Risk information influences risk behavior through consumers’ risk acceptance, and the provision of risk information and education are antecedent variables that cause lower-risk public behavior [16]. The more risk information the public has, the more powerful the effect is on risk perception [17]. Thus, risk information is a critical factor in managing public risk perception.

The risk perception of the public is affected by how risk information is presented and framed [8]. After researching the effects of public risk messages on risk perception, Williams and Noyes [9] reported that message content and message source reliability are associated with risk perception and emphasized that effective decision-making and safe behaviors require effective risk communication. It is crucial to inform people about risks by conveying risk analysis results [13,14].

According to Covello, von Winterfeldt, and Slovic [15], the main objectives of risk communication are information and education, behavior change and protective action, disaster warnings and emergency information, and joint problem solving and conflict resolution. In other words, provide people with information about risk and risk assessment, educate them, and encourage risk-reduction behavior, safety, and self-protection, then provide direction and behavioral guidance about risks, and involve the public in decision-making and risk management resolution. In order to achieve those objectives, information must be communicated to the public to ensure that they can perform safe behaviors correctly in risky situations. Providing the public with risk and control information is the correct risk communication method [16].

However, studies of consumers’ safety perception have thus far focused mostly on behavioral change through the provision of information [12,15], the influence of risk information on consumers’ risk perception [9,17,18,19,20], and the influence of individual factors constituting risk characteristics on risk perception [2,3,4,5,6,7,8,9,10,11,12,13,14,15,16,17,18,19,20,21,22,23,24,25,26,27,28,29]. No previous study has comprehensively analyzed the effects of risk characteristics on consumers’ risk perception dependent on information about how to deal with risks as a component of information provision. In order to protect consumers in the EU, Vincze et. al. [30] conducted a comprehensive risk assessment analysis associated with establishing a prompt alarm system to prevent or restrict the sale and use of products (e.g., tainted or harmful toys, cosmetics, and chemicals) that pose a serious risk to consumer health and safety. In addition, many studies conducted surveys on consumers’ risk perceptions and attitudes toward hazardous products [31,32,33,34]. Those studies found that consumers’ risk perception of hazardous products affected their individual risk management in terms of the environment and health as well as their attitude. Along with experts’ opinions, the public’s risk perception is an essential element in establishing theoretical systems and making decisions on public health and environmental policies [35,36,37,38,39,40,41]. Accordingly, surveys on consumers’ risk perception are necessary for the sustainable improvement of public health and environmental protection. 

Therefore, in this study of adult consumers in South Korea with a high level of anxiety and fear about biochemical products, we provided participants with information about how to use chemical products safely and examined differences in their risk perception. Our study aims to examine the following three topics: (1) Differences in the recognition of risk about chemical products for different demographics; (2) differences in risk characteristics and risk perception by the provision of safe risk management methods, and (3), differences in factors that affect risk perception through the provision of safe risk management methods. The findings of this study can be used as basic data when exploring methods to understand consumers’ perception of specific risks and managing risk perception.

## 2. Materials and Methods 

This study was conducted to investigate factors influencing the risk perception of consumers for chemical products. We conducted a survey for 10 days beginning 1 December 2018 through the online survey company MacromilEmbrain. The participants were 600 adult Korean consumers of chemical products. Population proportional allocation was made by considering gender, age, and place of residence.

We provided risk information sequentially to the respondents and asked them to answer twice. The first response was requested after providing hazard evaluation information about the target of risks, that is, chemical products. The second response was requested after providing information about how to use chemical products safely.

Data collection was based on a five-point Likert scale ranging from one point (strongly disagree) to five points (strongly agree), which comprised the theoretical concepts presented in Table 1 and described below. All reliability values (Cronbach’s alpha) were above the recommended critical point (0.70).

Data analyses were performed using SPSS 20.0 (IBM SPSS Inc., Chicago, IL, USA). Participants’ general characteristics, risk characteristics scores, and risk perception were analyzed using descriptive statistics such as frequency, percentage, mean, and standard deviation. Differences in risk perception according to demographic characteristics were analyzed with independent *t*-test and one-way ANOVA. Differences in risk characteristics and risk perception, depending on whether or not the risk response guideline information was provided, were analyzed with paired *t*-test. To identify the factors influencing risk perception, depending on whether or not safe usage information was provided, a multiple linear regression analysis was conducted.

Risk perception is an individual’s evaluation of the risk levels of specific objects [9]. According to psychometrics, experts define risks in a technical manner, whereas the general public evaluates risks by perceiving catastrophic potential and controllability in an integrated manner. To understand risk perception, trust should be considered [42]; other important influencing factors are control, familiarity, dread, and diffusion in time and space [21]. Risk perception is affected by risk characteristics such as familiarity, scientific knowledge, fear, catastrophic potential, the risk for future generations, and controllability [22,43]. Social and institutional trust, as well as risk characteristics, influence risk perception [44]. 

Human risk perceptions are divided into emotional and analytical risk perceptions [23]. Emotional risk perception is the evaluation of risks based on instinct and intuition, whereas analytical risk perception is the evaluation of risks based on logic, reason, and scientific deliberation. To summarize previous studies, risk perception is influenced by trust as well as the risk characteristics of catastrophic potential, controllability, familiarity, fear, knowledge, and the risk for future generations.

### 2.1. Catastrophic Potential

High catastrophic potential and risk experience are associated with a high degree of risk perception [45]. High catastrophic potential is sometimes perceived as very dangerous and dramatic in people’s judgment [24,25]. Furthermore, the perception of catastrophic potential is sometimes affected by related incidents [46].

**H1:** 
*The larger the catastrophic potential of a risk, the higher the risk perception.*


### 2.2. Controllability

People like to feel they can influence risks by taking measures to reduce them or completely prevent negative outcomes [47]. When people are in control of or have a choice to accept or avoid risks, they perceive risks as low [48]. In other words, risks are perceived as low when consumers believe they can control them on their own. 

In research related to ecological crisis caused by climate change, McDaniels et al. [49] found that controllability showed the highest correlation with risk perception because the greater the controllability, the lower the risk perception. Consequently, McDaniels et al. [49] strongly advocated the need for measures to regulate controllability in matters related to the ecological crisis. DeJoy [50] found a similar result in a study of traffic accidents: the higher the controllability of risks, the lower the risk perception. Related research findings can be also found in [51,52].

**H2:** 
*The greater the controllability of a risk, the lower the risk perception.*


### 2.3. Familiarity

Familiarity influences intuitive judgment of risks [26]. In particular, the more familiar the terms are, the lower the perceived risk. The more familiar the terms used for products, the lower the risk perceived by consumers [26,27]. Fischer and Frewer [53] conducted a study on risk perception for foods and found that the higher the familiarity with food hazards, the lower the measure of hazard perception. Smith et al. [54] conducted quantitative analysis on disaster rescue squad staff and found that the more familiar the disaster, the lower their risk perception.

**H3:** 
*The more familiar the risk, the lower the risk perception.*


### 2.4. Fear

Emotions influence individuals’ cognitive evaluations [55]. In the risk-as-feeling approach, emotions such as worry play a key role in risk judgment [55,56]. This approach categorizes emotions into two types. Anticipatory emotions are instinctive responses to risks, such as worry, fear, anxiety, and dread. Anticipated emotions are feelings a consumer expects to experience as a result of a decision, including integral or instinctive emotions resulting from the consumer’s selection itself and incidental emotions caused by other factors such as surrounding atmosphere and mood [57]. Consumers’ risk perceptions are directly associated with anticipatory emotions [28].

**H4:** 
*The greater the fear about risks, the higher the risk perception.*


### 2.5. Knowledge

Zhang et al. [58] conducted a study on risk perception of influenza according to the knowledge level of nurses and found that nurses with a high level of knowledge had lower risk perception and performed fewer vaccinations. In a study on the risk perception of nuclear power plant employees, Sjöberg and Drottz-Sjöberg [59] revealed that workers who knew more about radiation had higher knowledge about risks and higher risk perceptions. Gstraunthaler and Day [60] conducted a study on risk perception of avian influenza and found that the higher the scientific knowledge, the higher the risk perception.

**H5:** 
*The higher the scientific knowledge about risks, the higher the risk perception.*


### 2.6. Risk for Future Generations

The higher the influence a risk is likely to have on future generations, the higher the risk perception [61,62,63,64]. Many studies on risk factors that have a high influence on future generations involve environmental crises. Frischknecht [65] analyzed two different risk mindsets towards recycling approaches and found that concern for future generations resulted in risk aversion.

**H6:** 
*The greater the influence of risks on future generations, the higher the risk perception.*


## 3. Results

### 3.1. Samples and Measure

A total of 600 participants were selected using the proportional allocation sampling method based on population, considering gender, age, and areas of residence. The sample participants consisted of 51.3% males and 48.7% females. In terms of age, 22.3% were in their 20s, 23.7% in their 30s, 27.0% in their 40s, and 27.0% were in their 50s and older. In terms of education levels, 9.8% held a master’s degree or higher, 55.5% had bachelor’s degrees, 16.5% had vocational school level education, and 18.2% had a high school education or less.

### 3.2. Differences in Risk Perception according to Demographic Characteristics

Table 2 outlines the results of analyzing the differences in risk perception according to demographic characteristics. Women showed higher levels of risk perception than men regardless of the provision of information about safe usage. People in their 50s showed the highest risk perception when safe usage information was not provided, followed by those in their 30s and 20s. When safe usage information was provided, people older than 30 years showed the same level of risk perception, but those in their 20s showed a lower level of risk perception compared to other age groups. Married respondents showed higher levels of risk perception than singles regardless of the provision of safe usage information.

### 3.3. Differences in Risk Characteristics/Perception Based on Provision of Risk Response Guideline Information 

Table 3 shows the differences in risk characteristics and risk perception depending on whether or not safe usage information is provided. The means of catastrophic potential, controllability, fear, and scientific knowledge were higher when the safe usage information was provided than otherwise. In contrast, the level of risk for future generations was statistically significantly lower after providing the information. Only familiarity showed no difference in the mean value depending on whether or not the information was provided.

### 3.4. Factors Influencing Risk Perception Based on Provision of Safe Usage Information

Table 4 shows the result of analyzing the factors related to the influence of risk characteristics on risk perception. First, when safe usage information was not provided, catastrophic potential, fear, and the risk for future generations had positive effects, whereas familiarity had negative effects. In other words, the higher the perceived effects of chemical products on catastrophic potential, fear, and the risk for future generations, the higher the perception of risk. However, if the chemical products were considered more familiar, the effects of scientific knowledge, controllability, and demographic characteristics were less statistically significant.

When safe usage information was provided, catastrophic potential, fear, scientific knowledge, the risk for future generations, and age had positive effects on risk perception, but familiarity had a negative effect. When safe usage information was not provided, scientific knowledge and age had statistically significant effects, and the higher the knowledge and age, the higher the perception of risk became.

To examine the influence through standardized regression coefficients (beta value) depending on the provision of safe usage information, the influences of catastrophic potential, familiarity, and risk for future generation increased, whereas the influence of fear decreased.

## 4. Discussion

Our study investigated Korean consumers’ risk perception levels of chemical products and the factors influencing risk perception, depending on whether or not risk response guideline information was provided. Women had higher levels of risk perception about chemical products than men, and people in their 20s showed lower levels of risk perception compared to other age groups. Furthermore, married respondents showed higher risk perceptions than single people did, perhaps because the main users of chemical products in South Korea are housewives [66].

When information about safe usage of chemical products was provided, fear, catastrophic potential, scientific knowledge, and controllability, as well as risk perception, increased significantly. According to Nabi’s cognitive-functional model [67], when people feel that their physical safety is threatened, fear is induced, which motivates them to employ protective measures. Thus, risk perception has the positive effect of increasing protective behavior. In this respect, providing additional information about how to safely use the subject of risk has a positive effect on consumer safety.

When only the risk diagnosis information about chemical products was provided, there was a positive effect on catastrophic potential, fear, and the risk for future generations, but a negative effect on familiarity. When the information about how to use chemical products safely was provided, age and scientific knowledge additionally had positive effects. Furthermore, though the influence of fear decreased, the influences of catastrophic potential, familiarity, and the risk for future generations increased. These research findings indicate the importance of providing consumers with risk diagnosis information and risk response guidelines. However, while recent reports in the Korean media on safety accidents caused by household chemical products all contained risk diagnosis information, only 15.6% of reports provided risk response guidelines [68]. Son [68] suggested that providing consumers with risk response guidelines is effective in improving consumers’ risk perception and reducing anxiety. Providing consumers with risk response guidelines also has a positive effect on increasing rational safety behaviors [69].

Risk controllability, which is the ability to control risks, did not influence risk perception, perhaps because risk perception is a subjective assessment of risk and is affected by psychological factors. Our results suggest that providing information about how to avoid or deal with risks had positive effects on consumers’ risk perception. Risk perception acts positively or negatively depending on the subject of risk. As mentioned above, risk perception motivates consumers to protect themselves from risks on their own. However, excessive consumer risk perception can sometimes cause social problems [70]. Therefore, this study presents some implications in relation to the provision of safe usage information regarding subjects of risk. 

First, in order to manage the risk perception of consumers, information about how to safely handle potentially dangerous products (that is, information about how to avoid risks) should be provided. There are some problems in the contents of risk information currently provided in Korea. In the three major news outlets in Korea between January 2016 and December 2018, articles about safety problems of chemical products all contained diagnostic information that assessed risk, but only 15.6% provided information about safe risk management methods [71]. Social amplification of risk may occur along with excessive risk awareness in individuals if media reports on risk assessment information, rather than risk countermeasure information, persist [70]. Therefore, caution must be taken not to provide only risk assessment information; safe risk management methods must also be provided. Particularly if consumers ignore or fail to perceive specific risks, safe risk management methods should be provided continuously to raise risk perception because when information about safe risk management methods is provided, consumers’ evaluation of risk characteristics as well as risk perception becomes higher. Information and education should be provided particularly to young consumers in their 20s, who have low risk perception about chemical products. Providing consumer education should be considered because consumers’ risk perception affects public health and personal risk management [30,31,32,33,34]; along with expert opinions, the public’s risk perception is an essential element in improving public health and the policy-making process [35,36,37,38,39,40].

Second, various antecedent variables affecting risks must be considered to manage risk perception. Among the factors influencing Korean consumers’ risk perception, fear and the risk for future generations, which are psychological factors, showed relatively higher influence. In contrast, the influence of scientific knowledge, which is a cognitive factor, showed the lowest influence. Therefore, for management of risk perception, individual or social measures to ensure the safety of future generations must be provided while managing psychological factors such as fear.

Risk perception is an important antecedent variable for behavior intention [72]. Consumers’ risk perception of product accidents affects their use of the product [73]. Consumers’ excessive risk perception may lead to social amplification [48]. For instance, the 2008 protest against resuming U.S. beef imports to Korea was triggered by consumers’ fear of mad cow disease due to their ignorance of risk management. This resulted in excessive risk perception in Korean consumers, setting off the massive civil movement called the Candlelight Rally [74]. The 2008 protest was indicative of the structural tradeoff in providing risk information [75]. Decision-making with no consideration of risk perception has serious repercussions, as risk perception is an important factor influencing consumer decision-making [35]. Consumers’ risk perception induces self-protective behavior, but in excess, it can result in social problems. Accordingly, consumers’ risk perception should be managed at the public level.

By analyzing consumers’ risk perception about chemical products, our research proposes methods to understand consumer perceptions and promote consumers’ risk perception appropriately according to the situation. However, our study has some limitations. First, we investigated risk characteristics and risk perception only for chemical products; consumers’ perception needs to be investigated in other situations as well. Second, this study only examined the effects of demographic characteristics and risk characteristics on risk perception. Therefore, it will be necessary to discover other factors that affect risk perception.

## 5. Conclusions

Our study showed that risk management methods must be provided along with risk diagnostic information when providing risk information about products to consumers. This study made clear that the content of risk information is also a factor that significantly affects consumers’ safety behavior. Therefore, it is suggested that items requiring the provision of risk management information to consumers should be added to the guidelines of official government documents and articles in journalism. In addition, to promote consumer safety behaviors consistently, educational efforts and campaigns must provide risk response guidelines related to consumer safety issues.

## Figures and Tables

**Table 1 ijerph-17-01207-t001:** Research concepts: risk characteristics and perceptions.

Concept		Statements Measuring the Concept	Reliability (Cronbach’s Alpha)
Risk Characteristics	Catastrophic potential	I think big accidents can occur by using chemical products such as disinfectants and detergents.	0.837
	Controllability	I think I can protect myself from the risks of chemical products if I make an effort.	
	Familiarity	I think accidents caused by chemical products such as disinfectants and detergents are common risks.	
	Fear	I am afraid that I may be endangered by using chemical products such as disinfectants and detergents.	
	Knowledge	I know many things about the risks of chemical products such as disinfectants and detergents.	
	Risk for future generations	I think that using chemical products such as disinfectants and detergents will have negative effects on future generations.	
Risk Perception		I think that using chemical products such as disinfectants and detergents is dangerous.	-

**Table 2 ijerph-17-01207-t002:** Differences in risk perception according to demographic characteristics.

Categories		N	Information Was Not Provided			Information Was Provided		
			Mean	S.D	t/F-value	Mean	S.D	t/F-value
Sex	Men	308	3.555	0.869	−1.419	3.575	0.810	−3.357**
	Women	292	3.651	0.779		3.791	0.769	
Age	20s	134	3.194 c	0.863	19.078***	3.3731 b	0.856	11.673***
	30s	142	3.549 b	0.813		3.6549 a	0.817	
	40s	162	3.728 ab	0.764		3.7346 a	0.762	
	50s	162	3.858 a	0.738		3.9012 a	0.680	
Marriage	Married	349	3.7937	0.749	6.815***	3.8252	0.732	5.264***
	Single	251	3.3347	0.858		3.4781	0.841	

** *p* < 0.01, *** *p* < 0.001.

**Table 3 ijerph-17-01207-t003:** Differences in risk characteristics/perception based on the provision of risk response guideline information.

Categories		Information Was Not Provided		Information Was Provided		t-value
		Mean	Standard Deviation	Mean	Standard Deviation	
Risk characteristics	Catastrophic potential	3.510	0.863	3.590	0.835	−2.664**
	Controllability	3.322	0.887	3.550	0.789	−6.588***
	Familiarity	2.253	0.772	2.298	0.751	−1.457
	Fear	3.493	0.903	3.607	0.876	−4.073***
	Scientific knowledge	2.823	0.835	2.915	0.848	−2.905**
	Risk for future generations	3.963	0.739	3.883	0.751	2.897**
Risk perception		3.602	0.827	3.680	0.797	−2.628**

** *p* < 0.01, *** *p* < 0.001.

**Table 4 ijerph-17-01207-t004:** Factors influencing risk perception based on the provision of safe usage information.

Categories	Information Was Not Provided				Information Was Provided			
	B	Standard error	Beta	t-value	B	Standard error	Beta	t-value
Constant	0.850	0.254		3.351**	0.697	0.247		2.818**
Age	0.004	0.002	0.053	1.608	0.009	0.002	0.117	3.549***
Sex	0.045	0.043	0.028	1.045	−0.030	0.045	−0.018	−0.665
Marriage	0.022	0.053	0.014	0.419	−0.016	0.055	−0.009	−0.281
Monthly average household income	0.000	0.000	0.046	1.710	0.000	0.000	0.007	0.263
Level of education	−0.013	0.024	−0.015	−0.563	0.007	0.025	0.007	0.264
Catastrophic potential	0.150	0.037	0.157	4.061***	0.156	0.035	0.163	4.464***
Controllability	−0.006	0.028	−0.006	−0.226	0.017	0.026	0.018	0.638
Familiarity	−0.103	0.035	−0.097	−2.978**	−0.158	0.034	−0.148	−4.646***
Fear	0.361	0.037	0.398	9.882***	0.307	0.035	0.327	8.704***
Scientific knowledge	0.051	0.027	0.054	1.904	0.059	0.029	0.059	2.028*
Risk for future generations	0.232	0.039	0.219	5.924***	0.261	0.038	0.233	6.840***
F-value	78.236***				78.540***			
$R^{2}$	0.594				0.595			
Adjusted $R^{2}$	0.586				0.587			

* *p* < 0.05, ** *p* < 0.01, *** *p* < 0.001.

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
