# Peer review of "Korean Consumers’ Recognition of Risks Depending on the Provision of Safety Information for Chemical Products"

_ijerph, 2020, doi:10.3390/ijerph17041207_

Round 1
Reviewer 1 Report
I found the subject of the MS interesting and new. It was nively written with an appropriate length. However, I failed to see the relevance of the results: What new this study brings and where these results can be applied? This need to be added to results and conslusions. I´d would have liked to see an example about what problems or hazards there currently are. Please add a case to clarify the meaning of this study.

Author Response
Comment 1
|
Reviewers’ comment: overall comment: What new this study brings and where these results can be applied? This need to be added to results and conclusions |
Answer: In response to the reviewers' comments, we added information about the new results from this study and their applications to the Results and and Conclusions sections, lines 318-324:
Our study showed that risk management methods must be provided along with risk diagnostic information when providing risk information about products to consumers. This study made clear that the content of risk information is also a factor that significantly affects consumers' safety behavior. Therefore, it is suggested that items requiring the provision of risk management information to consumers should be added to the guidelines of official government documents and articles in journalism. In addition, to promote consumers’ safety behaviors consistently, educational efforts and campaigns must provide risk response guidelines related to consumers’ safety issues.
□ Comment 2
|
Reviewers’ comment: overall comment: I´d would have liked to see an example about what problems or hazards there currently are.Please add a case to clarify the meaning of this study. |
Answer: To understand today’s issues and risks, we added to the Discussion section risk information issues that arose in the past in Korea and current issues on the provision of risk information. Please see lines 260-266 and 299-309:
These research findings indicate how important it is to provide consumers with risk diagnosis information and risk response guidelines. However, while recent reports in the Korean media on safety accidents caused by household chemical products all contained risk diagnosis information, only 15.6% of reports provided risk response guidelines [68]. Son [68] suggested that providing consumers with risk response guidelines is effective in improving consumers’ risk perception and reducing anxiety. Providing consumers with risk response guidelines also causes a positive effect on increasing rational safety behaviors [69].
Risk perception is an important antecedent variable for behavior intention [72]. Consumers’ risk perception about product accidents affects their use of the product [73]. Consumers’ excessive risk perception may lead to social amplification [48]. For instance, the 2008 protest against resuming U.S. beef imports to Korea was triggered by consumers’ fear of mad cow disease due to their ignorance of risk management. This resulted in excessive risk perception in Korean consumers, setting off the massive civil movement called the Candlelight Rally [74]. The 2008 protest was indicative of the structural tradeoff in providing risk information [75]. Decision-making with no consideration of risk perception has serious repercussions, as risk perception is an important factor influencing consumers’ decision-making [35]. Consumers’ risk perception induces self-protective behavior, but in excess it can result in social problems. Accordingly, consumers’ risk perception should be managed at the public level.
+++References
Li, Z. Valuing acute health risks of air pollution in the Jinchuan Mining Area, China: a choice experiment with perceived exposure and hazardousness as co-determinants. Int J Environ Res Public Health 2019, 16(22), 4563. Keeney, R.L. Understanding life‐threatening risks. Risk Anal 1995, 15(6), 627–37. Song, E. A study on consumers’ risk perception and safety-seeking behavior according to risk information frame and provision method—focused on household chemical products 2019, 15(4), 123-144. Noar, S.M.; Zimmerman, R.S. Health behavior theory and cumulative knowledge regarding health behaviors: are we moving in the right direction? Health Educ Res 2005, 20(3), 275–290. Weegels, M.F.; Hanis, H. Risk perception in consumer product use. Accident Anal Prev 2000, 32(3), 365-370. Noh, J.C. Mad Cow Disease fears and crisis communication of the unknown in the 2008 Candlelight Vigils. Econ Soc 2009, 84, 158-182. Youn, S.M.; Cho, K.M. The problems of science communication in Korea: focusing on the “Mad Cow Disease” controversy. The Korean Journal for the History of Science 2011, 33, 75-117.
□ Comment 3
|
# Risk information influences risk behavior through consumers’ risk acceptance, and the provision of risk information and education are antecedent variables that cause lower-risk public behavior →Reviewers’ comment: Memo4(line 53) This is an important but difficult subject to handle. One should not overestimate the ability of a typical consumer to evaluate the risks of products and their ingredients. |
Answer: We thank the reviewer for this observation, and we agree. We reviewed the literature and introduced previous findings on the effect of risk information on consumers to evaluate the risk of products. Please see the Introduction, lines 52-57:
Unlike experts who assess risks based on scientifically proven facts, consumers assess risks subjectively through their own experience, knowledge, and risk information gained from various sources [10]. Safety information interventions change consumers’ attitudes toward safe behaviors as well as their actual behaviors, thus motivating them to process and use health information. Consumers’ safety assessments vary depending on the quality of information [11–12].
+++References
Nauta, M. J.; Fischer, A.R.H.; van Asselt, E. D.; de Jong, A. E. I.; Frewer, L. J.; de Jonge, R. Food safety in the domestic environment: the effect of consumer risk information on human disease risks. Risk Anal 2008, 28(1), 179-193. Büchter, R. B.; Fechtelpeter, D.; Knelangen, M.; Ehrlich, M.; Waltering, A. Words or numbers? Communicating risk of adverse effects in written consumer health information: a systematic review and meta-analysis. BMC Med Inform Decis 2014, 14, 76.
□ Comment 4
|
#2.1Theoretical Concepts →Reviewers’ comment: Memo6(line 119) Consider writing full sentences in a chapter instead. |
Answer: All the text in 2.1 Theoretical Concepts, which had been written in abbreviated forms, was changed to full sentences, starting in line 140:
2.1.1 Catastrophic Potential
High catastrophic potential and risk experience are associated with a high degree of risk perception [45]. High catastrophic potential is sometimes perceived as very dangerous and dramatic in people’s judgment [24,25]. Furthermore, perception about catastrophic potential is sometimes affected by related incidents [46].
H1: The larger the catastrophic potential of a risk, the higher the risk perception.
2.1.2 Controllability
People like to feel they can influence risks by taking measures to reduce them or completely prevent negative outcomes [47]. When people are in control of or have a choice to accept or avoid risks, they perceive risks as low [48]. In other words, risks are perceived as low when consumers believe they can control them on their own.
In research related to ecological crisis caused by climate change, McDaniels et al. [49] found that controllability showed the highest correlation with risk perception, because the greater the controllability, the lower the risk perception. Consequently, McDaniels et al. [49] strongly advocated the need for measures to regulate controllability in matters related to ecological crisis. DeJoy [50] found a similar result in a study of traffic accidents: the higher the controllability of risks, the lower the risk perception. Related research findings can be also found in [51-52].
H2: The greater the controllability of a risk, the lower the risk perception.
□ Comment 5
|
#개조식으로 정리된 2.1Theoretical Concepts중 dots가생략됨 →Reviewers’ comment: Memo7(line 119) Add dots here as well or remove all. |
Answer: Following the reviewers' Comment 3, the entire section was modified to include full sentences, and bullet points were deleted in the process. Thank you.
□ Comment 6
|
#In the final sample N=600, · gender: 51.3% male, 48.7% female · age range: 22.3% in their 20s, 23.7% in their 30s, 27.0% in their 40s, and 27.0% over 50 · education level: 9.8% master’s degree or higher, 55.5% bachelor’s degree, 16.5% some college, and 18.2% high school or less →Reviewers’ comment: Write full sentences! |
Answer: Following the reviewers' comments, the descriptions of the samples were modified to full sentences in lines 196-202:
A total of 600 participants were selected using the proportional allocation sampling method based on population, considering gender, age, and areas of residence. The sample participants consisted of 51.3% males and 48.7% females. In terms of age, 22.3% were in their 20s, 23.7% in their 30s, 27.0% in their 40s, and 27.0% were in their 50s and older. In terms of education levels, 9.8% held a master’s degree or higher, 55.5% had bachelor’s degrees, 16.5% had vocational school level education, and 18.2% had a high school education or less.
□ Comment 7
|
Reviewers’ comment: 수치표시방법, 오탈자수정, 참고문헌작성방법등 |
Answer: Thank you for your highly detailed reviews. We revised wording in the text as shown in the table below, according to the reviewers' comments. Also, overall edits, typographical errors, numerical formats, bibliography formats and others were examined and modified accordingly. You can find the details both in the table below and the text.
|
Division |
No. of line |
Request of revision |
Before revision |
After revision |
|
Memo 1 |
22 |
.586 |
.586 |
0.586 |
|
Memo 2 |
28 |
Change this to a more specific word |
fear |
chemophobia |
|
Memo 3 |
29 |
Use ; between keywords |
Used ‘,’ |
Use ‘ ; ‘ |
|
Memo 5 |
94 |
Use 0.70 for clarity. |
.70 |
0.70 |
|
Memo 9 |
222 |
Please write numbers, e.g. 0.850, throughout the MS. |
- |
All numbers are written as 0.xxx. |
|
Memo 10 |
303 |
Typographical error correction |
conflict |
conflicts |
|
Memo 11 |
312 |
Bibliography format modification Used ; between authors |
- |
The journal template specifies using semicolons between authors’ names (and commas to separate last names from first initials). This is now consistent throughout the References section. |
|
Memo 12-13 |
329,331 |
Abbreviation? |
- |
All citations were checked and revised |
|
Memo 14-16 |
351, 374, 389 |
Issue? |
|
All citations were checked and revised |
++We appreciate the excellent comments of the reviewers, which helped us improve the quality of our manuscript. Thank you! ++

Reviewer 2 Report
The authors have studied Korean Consumers’ Recognition of Risks Depending on the Provision of Safe Information for Chemical Products. They conducted an online survey for 10 days and got 600 samples. The experimental design is arbitrary and should be more logical and scientific, but I just do not see anything novel in this work. We can't see how the questionnaire and survey methods have improved compared with the previous work. This is an investigation report, not a scientific paper, so I do not warrant publication in International Journal of Environmental Research and Public Health. In my opinion, the journal should only publish manuscripts that have enough data and answer a scientific question that has previously not had an answer. I just do not see what that previously unanswered question would be in this case, and so I recommend major revision of the manuscript.
Author Response
Comments from Reviewer 2 and Revisions
□ Comment 1
|
Reviewers’ comment:The experimental design is arbitrary and should be more logical and scientific, but I just do not see anything novel in this work. We can't see how the questionnaire and survey methods have improved compared with the previous work. |
Answer: We added information to the Introduction about the differences between previous studies and this study about public health and consumers’ perception. Please see lines 50 through 57:
Through information and communication technology, the general public receives information in a variety of ways, but people evaluate the same risks differently depending on the amount and content of risk information conveyed through the media. Unlike experts who assess risks based on scientifically proven facts, consumers assess risks subjectively through their own experience, knowledge, and risk information gained from various sources [10]. Safety information interventions change consumers’ attitudes toward safe behaviors as well as their actual behaviors, thus motivating them to process and use health information. Consumers’ safety assessments vary depending on the quality of information [11–12]. Consumer information affects consumers’ perceptions, and newly provided information changes their perceptions [13-15]. Risk information influences risk behavior through consumers’ risk acceptance, and the provision of risk information and education are antecedent variables that cause lower-risk public behavior [16]. The more risk information the public has, the more powerful the effect is on risk perception [17]. Thus, risk information is a critical factor in managing public risk perception.
+++References
Kim, K. H.; Yoo, H. J.; Song, E. An analysis on the structural model for consumer trust-anxiety-competency by source of information—focused on chemical household products. Crisisonomy 2017, 13(3), 141-158. Nauta, M. J.; Fischer, A.R.H.; van Asselt, E. D.; de Jong, A. E. I.; Frewer, L. J.; de Jonge, R. Food safety in the domestic environment: the effect of consumer risk information on human disease risks. Risk Anal 2008, 28(1), 179-193. Büchter, R. B.; Fechtelpeter, D.; Knelangen, M.; Ehrlich, M.; Waltering, A. Words or numbers? Communicating risk of adverse effects in written consumer health information: a systematic review and meta-analysis. BMC Med Inform Decis 2014, 14, 76. Fischhoff, B. Risk perception and communication unplugged: twenty years of process. Risk Anal 1995, 15(2), 137–145. Keeney, R.L.; von Winterfeldt, D. Improving risk communication. Risk Anal 1986, 6(4), 417–424. Covello, V.T.; von Winterfeldt, D.; Slovic, P. Risk communication: a review of the literature. Risk Abstracts 1986, 3, 171-182. Hath, R.; Nathan, K. Public relation’s role in risk communication; information, rhetoric and power. Public Relations Quarterly 1991, 35(4), 15-22.
□ Comment 2
|
Reviewers’ comment:This is an investigation report, not a scientific paper, so I do not warrant publication in International Journal of Environmental Research and Public Health. |
Answer: Some previous studies on risk perception of household chemicals, which we thought are relevant to this journal, were added to supplement our findings. Please see lines 77 through 92.
However, studies on consumers’ safety perception have thus far focused mostly on behavioral change through the provision of information [12,15], the influence of risk information on consumers’ risk perception [9, 17–20], and the influence of individual factors constituting risk characteristics on risk perception [28–29]. No previous study comprehensively analyzed the effects of risk characteristics on consumers’ risk perception dependent on information about how to deal with risks as a component of information provision. In order to protect consumers in the EU, Vincze, et. al. [30] conducted a comprehensive risk assessment analysis associated with establishing a prompt alarm system to prevent or restrict the sale and use of products (e.g., tainted or harmful toys, cosmetics, and chemicals) that pose a serious risk to consumers’ health and safety. In addition, many studies conducted surveys on consumers’ risk perception and attitudes toward hazardous products [31–34]. Those studies found that consumers’ risk perception about hazardous products affected their individual risk management in terms of environment and health as well as their attitude. Along with experts’ opinions, the public’s risk perception is an essential element in establishing theoretical systems and making decisions on public health and environmental policies [35–41]. Accordingly, surveys on consumers’ risk perception are necessary for the sustainable improvement of public health and environmental protection.
+++References
Williams, D.J.; Noyes, J.M. How does our perception of risk influence decision-making? Implications for the design of risk information. Theor Issues Ergon Sci 2007, 8(1), 1-35. Büchter, R. B.; Fechtelpeter, D.; Knelangen, M.; Ehrlich, M.; Waltering, A. Words or numbers? Communicating risk of adverse effects in written consumer health information: a systematic review and meta-analysis. BMC Med Inform Decis 2014, 14, 76. Covello, V.T.; von Winterfeldt, D.; Slovic, P. Risk communication: a review of the literature. Risk Abstracts 1986, 3, 171-182. Kummeneje, A.M.; Rundmo, T. Risk perception, worry, and pedestrian behaviour in the Norwegian population. Accident Anal Prev 2019, 133(December), 1-9. Hidaka, T.; Kakamu, T.; Endo, S.; Kasuga, H.; Masuishi, Y.; Kumagai, T.; Fukushima, T. Association of anxiety over radiation exposure and cquisition of knowledge regarding occupational health management in operation leader candidates of radioactivity decontamination workers in fukushima, japan: across-sectional study. Int J Environ Res Public Health 2020, 17(1), 228. Vincze, S.; Al Dahouk, S.; Dieckmann, R. Microbiological safety of non-food products: what can we learn from the RAPEX database? Int J Environ Res Public Health 2019, 16(9), 1599.[31] Yang, B.; Owusu, D.; Popova, L. Effects of a nicotine fact sheet on perceived risk of nicotine and e-cigarettes and intentions to seek information about and use e-cigarettes. Int J Environ Res Public Health 2020, 17(1), 131.
Jin, H. J.; Han, D. H. College students’ experience of a food safety class and their responses to the MSG issue. Int J Environ Res Public Health 2019, 16(16), 2977. Shan, L.; Wang, S.; Wu, L.; Tsai, F. S. Cognitive biases of consumers’ risk perception of foodborne diseases in China: examining anchoring effect. Int J Environ Res Public Health 2019, 16(13), 2268. Cembalo, L.; Caso, D.; Carfora, V.; Caracciolo, F.; Lombardi, A.; Cicia, G. The “Land of Fires” toxic waste scandal and its effect on consumer food choices. Int J Environ Res Public Health 2019, 16(1), 165. Li, Z. Valuing acute health risks of air pollution in the Jinchuan Mining Area, China: a choice experiment with perceived exposure and hazardousness as co-determinants. Int J Environ Res Public Health 2019, 16(22), 4563.
++We appreciate the excellent comments of the reviewers, which helped us improve the quality of our manuscript. Thank you! ++
Reviewer 3 Report
presented manuscript is interesting, but I am not sure if it fits well within the scope of the journal. It is correct but if authors could underline, parhaps in the discussion part and in the Introduction part of the manuscript, the connection of their research with the influence on the environment it would significantly improve the quality of the paper
Author Response
Comments from Reviewer 3 and Revisions
□ Comment
|
Reviewers’ comment:presented manuscript is interesting, but I am not sure if it fits well within the scope of the journal. It is correct but if authors could underline, parhaps in the discussion part and in the Introduction part of the manuscript, the connection of their research with the influence on the environment it would significantly improve the quality of the paper. |
Answer: Previous studies on the risk perception of chemicals that we thought relevant to this journal were added to Introduction and Discussion sections. Please see lines 50-57, 77-92, 259-265, and 298-308.
Lines 50 through 57:
Through information and communication technology, the general public receives information in a variety of ways, but people evaluate the same risks differently depending on the amount and content of risk information conveyed through the media. Unlike experts who assess risks based on scientifically proven facts, consumers assess risks subjectively through their own experience, knowledge, and risk information gained from various sources [10]. Safety information interventions change consumers’ attitudes toward safe behaviors as well as their actual behaviors, thus motivating them to process and use health information. Consumers’ safety assessments vary depending on the quality of information [11–12]. Consumer information affects consumers’ perceptions, and newly provided information changes their perceptions [13-15]. Risk information influences risk behavior through consumers’ risk acceptance, and the provision of risk information and education are antecedent variables that cause lower-risk public behavior [16]. The more risk information the public has, the more powerful the effect is on risk perception [17]. Thus, risk information is a critical factor in managing public risk perception.
+++References
Kim, K. H.; Yoo, H. J.; Song, E. An analysis on the structural model for consumer trust-anxiety-competency by source of information—focused on chemical household products. Crisisonomy 2017, 13(3), 141-158. Nauta, M. J.; Fischer, A.R.H.; van Asselt, E. D.; de Jong, A. E. I.; Frewer, L. J.; de Jonge, R. Food safety in the domestic environment: the effect of consumer risk information on human disease risks. Risk Anal 2008, 28(1), 179-193. Büchter, R. B.; Fechtelpeter, D.; Knelangen, M.; Ehrlich, M.; Waltering, A. Words or numbers? Communicating risk of adverse effects in written consumer health information: a systematic review and meta-analysis. BMC Med Inform Decis 2014, 14, 76. Fischhoff, B. Risk perception and communication unplugged: twenty years of process. Risk Anal 1995, 15(2), 137–145. Keeney, R.L.; von Winterfeldt, D. Improving risk communication. Risk Anal 1986, 6(4), 417–424. Covello, V.T.; von Winterfeldt, D.; Slovic, P. Risk communication: a review of the literature. Risk Abstracts 1986, 3, 171-182. Hath, R.; Nathan, K. Public relation’s role in risk communication; information, rhetoric and power. Public Relations Quarterly 1991, 35(4), 15-22.
Lines 77 through 92:
However, studies on consumers’ safety perception have thus far focused mostly on behavioral change through the provision of information [12,15], the influence of risk information on consumers’ risk perception [9, 17–20], and the influence of individual factors constituting risk characteristics on risk perception [28–29]. No previous study comprehensively analyzed the effects of risk characteristics on consumers’ risk perception dependent on information about how to deal with risks as a component of information provision. In order to protect consumers in the EU, Vincze, et. al. [30] conducted a comprehensive risk assessment analysis associated with establishing a prompt alarm system to prevent or restrict the sale and use of products (e.g., tainted or harmful toys, cosmetics, and chemicals) that pose a serious risk to consumers’ health and safety. In addition, many studies conducted surveys on consumers’ risk perception and attitudes toward hazardous products [31–34]. Those studies found that consumers’ risk perception about hazardous products affected their individual risk management in terms of environment and health as well as their attitude. Along with experts’ opinions, the public’s risk perception is an essential element in establishing theoretical systems and making decisions on public health and environmental policies [35–41]. Accordingly, surveys on consumers’ risk perception are necessary for the sustainable improvement of public health and environmental protection.
+++References
Williams, D.J.; Noyes, J.M. How does our perception of risk influence decision-making? Implications for the design of risk information. Theor Issues Ergon Sci 2007, 8(1), 1-35. Büchter, R. B.; Fechtelpeter, D.; Knelangen, M.; Ehrlich, M.; Waltering, A. Words or numbers? Communicating risk of adverse effects in written consumer health information: a systematic review and meta-analysis. BMC Med Inform Decis 2014, 14, 76. Covello, V.T.; von Winterfeldt, D.; Slovic, P. Risk communication: a review of the literature. Risk Abstracts 1986, 3, 171-182. Kummeneje, A.M.; Rundmo, T. Risk perception, worry, and pedestrian behaviour in the Norwegian population. Accident Anal Prev 2019, 133(December), 1-9. Hidaka, T.; Kakamu, T.; Endo, S.; Kasuga, H.; Masuishi, Y.; Kumagai, T.; Fukushima, T. Association of anxiety over radiation exposure and cquisition of knowledge regarding occupational health management in operation leader candidates of radioactivity decontamination workers in fukushima, japan: across-sectional study. Int J Environ Res Public Health 2020, 17(1), 228. Vincze, S.; Al Dahouk, S.; Dieckmann, R. Microbiological safety of non-food products: what can we learn from the RAPEX database? Int J Environ Res Public Health 2019, 16(9), 1599. Yang, B.; Owusu, D.; Popova, L. Effects of a nicotine fact sheet on perceived risk of nicotine and e-cigarettes and intentions to seek information about and use e-cigarettes. Int J Environ Res Public Health 2020, 17(1), 131. Jin, H. J.; Han, D. H. College students’ experience of a food safety class and their responses to the MSG issue. Int J Environ Res Public Health 2019, 16(16), 2977. Shan, L.; Wang, S.; Wu, L.; Tsai, F. S. Cognitive biases of consumers’ risk perception of foodborne diseases in China: examining anchoring effect. Int J Environ Res Public Health 2019, 16(13), 2268. Cembalo, L.; Caso, D.; Carfora, V.; Caracciolo, F.; Lombardi, A.; Cicia, G. The “Land of Fires” toxic waste scandal and its effect on consumer food choices. Int J Environ Res Public Health 2019, 16(1), 165. 35. Li, Z. Valuing acute health risks of air pollution in the Jinchuan Mining Area, China: a choice experiment with perceived exposure and hazardousness as co-determinants. Int J Environ Res Public Health 2019, 16(22),
Lines 259 through 265:
These research findings indicate how important it is to provide consumers with risk diagnosis information and risk response guidelines. However, while recent reports in the Korean media on safety accidents caused by household chemical products all contained risk diagnosis information, only 15.6% of reports provided risk response guidelines [68]. Son [68] suggested that providing consumers with risk response guidelines is effective in improving consumers’ risk perception and reducing anxiety. Providing consumers with risk response guidelines also causes a positive effect on increasing rational safety behaviors [69].
+++References
Song, E. A study on consumers’ risk perception and safety-seeking behavior according to risk information frame and provision method—focused on household chemical products 2019, 15(4), 123-144. Song, E. A study on risk perception, risk information acceptance, safety attitude, and safety actions of consumers according to the types of exposure to risk communication—based on the application of the consumers’ risk information acceptance models. 2019, 15(2), 49-70.
Lines 298 through 308:
Providing consumer education information should be considered because consumers’ risk perception affects public health and personal risk management [30–34]; along with expert opinions, the public’s risk perception is an essential element in improving public health and the policy-making process [35–40].
+++References
Vincze, S.; Al Dahouk, S.; Dieckmann, R. Microbiological safety of non-food products: what can we learn from the RAPEX database? Int J Environ Res Public Health 2019, 16(9), 1599. Yang, B.; Owusu, D.; Popova, L. Effects of a nicotine fact sheet on perceived risk of nicotine and e-cigarettes and intentions to seek information about and use e-cigarettes. Int J Environ Res Public Health 2020, 17(1), 131. Jin, H. J.; Han, D. H. College students’ experience of a food safety class and their responses to the MSG issue. Int J Environ Res Public Health 2019, 16(16), 2977. Shan, L.; Wang, S.; Wu, L.; Tsai, F. S. Cognitive biases of consumers’ risk perception of foodborne diseases in China: examining anchoring effect. Int J Environ Res Public Health 2019, 16(13), 2268. Cembalo, L.; Caso, D.; Carfora, V.; Caracciolo, F.; Lombardi, A.; Cicia, G. The “Land of Fires” toxic waste scandal and its effect on consumer food choices. Int J Environ Res Public Health 2019, 16(1), 165. Li, Z. Valuing acute health risks of air pollution in the Jinchuan Mining Area, China: a choice experiment with perceived exposure and hazardousness as co-determinants. Int J Environ Res Public Health 2019, 16(22), 4563. Karaye, I. M.; Horney, J. A.; Retchless, D. P.; Ross, A. D. Determinants of hurricane evacuation from a large representative sample of the US Gulf Coast. Int J Environ Res Public Health 2019, 16(21), 4268. Gao, S.; Li, W.; Ling, S.; Dou, X.; Liu, X. An empirical study on the influence path of environmental risk perception on behavioral responses in China. Int J Environ Res Public Health 2019, 16(16), 2856. Han, G.; Yan, S. Does food safety risk perception affect the public’s trust in their government? An empirical study on a national survey in China. Int J Environ Res Public Health 2019, 16(11), 1874. Shin, M.; Werner, A. K.; Strosnider, H.; Hines, L. B.; Balluz, L.; Yip, F. Y. Public perceptions of environmental public health risks in the United States. Int J Environ Res Public Health 2019, 16(6), 1045. Teysseire, R.; Lecourt, M.; Canet, J.; Manangama, G.; Sentilhes, L.; Delva, F. Perception of environmental risks and behavioral changes during pregnancy: a cross-sectional study of French postpartum women. Int J Environ Res Public Health 2019, 16(4), 565.
++We appreciate the excellent comments of the reviewers, which helped us improve the quality of our manuscript. Thank you! ++
Round 2
Reviewer 1 Report
The manuscript has greatly been improved after the changes and additions. Now it deserves to be published in IJERPH.
Author Response
Response to Comments from Reviewer 1 and Revisions(Round 2)
□ Comment1
|
Reviewers’ comment: The manuscript has greatly been improved after the changes and additions. Now it deserves to be published in IJERPH. |
Answer: We appreciate your review.
□ Comment2
|
Reviewers’ comment: English language and style are fine/minor spell check required |
Answer: We have checked manuscript and it has been reviewed by Editage, which is an English editing company. All the revised spellings have been highlighted.
We appreciate and thank the reviewers for their insightful comments, which helped us improve the quality of our manuscript.
